# Ni Oxidation State and Ligand Saturation Impact on the Capability of Octaazamacrocyclic Complexes to Bind and Reduce CO_2_

**DOI:** 10.3390/molecules26144139

**Published:** 2021-07-07

**Authors:** Barbora Vénosová, Ingrid Jelemenská, Jozef Kožíšek, Peter Rapta, Michal Zalibera, Michal Novotný, Vladimir B. Arion, Lukáš Bučinský

**Affiliations:** 1Faculty of Chemical and Food Technology, Institute of Physical Chemistry and Chemical Physics, Slovak University of Technology in Bratislava, Radlinského 9, 81237 Bratislava, Slovakia; barbora.venosova@stuba.sk (B.V.); ingrid.jelemenska@stuba.sk (I.J.); jozef.kozisek@stuba.sk (J.K.); peter.rapta@stuba.sk (P.R.); michal.zalibera@stuba.sk (M.Z.); 2Department of Chemistry, Faculty of Natural Sciences, Constantine the Philosopher University in Nitra, 94974 Nitra, Slovakia; 3Department of Physics, Faculty of Science, University of Ostrava, 30. dubna 22, 701 03 Ostrava, Czech Republic; michal.novotny@osu.cz; 4Institute of Inorganic Chemistry, University of Vienna, Währinger Strasse 42, 1090 Vienna, Austria; vladimir.arion@univie.ac.at

**Keywords:** DFT, nickel complexes, CO_2_ reduction, CO_2_ fixation, CO_2_ catalysis, domain averaged fermi holes, bonding analysis, QTAIM, MPA

## Abstract

Two 15-membered octaazamacrocyclic nickel(II) complexes are investigated by theoretical methods to shed light on their affinity forwards binding and reducing CO_2_. In the first complex ^1^[**Ni^II^L**]^0^, the octaazamacrocyclic ligand is grossly unsaturated (π-conjugated), while in the second ^1^[**Ni^II^LH**]^2+^ one, the macrocycle is saturated with hydrogens. One and two-electron reductions are described using Mulliken population analysis, quantum theory of atoms in molecules, localized orbitals, and domain averaged fermi holes, including the characterization of the Ni-C_CO2_ bond and the oxidation state of the central Ni atom. It was found that in the [**NiLH**] complex, the central atom is reduced to Ni^0^ and/or Ni^I^ and is thus able to bind CO_2_ via a single σ bond. In addition, the two-electron reduced ^3^[**NiL**]^2−^ species also shows an affinity forwards CO_2_.

## 1. Introduction

The catalytic reduction of carbon dioxide to C1 chemicals (or even to liquid fuels, e.g., methanol) is receiving considerable attention. Lowering the concentration of atmospheric CO_2_ and/or the control of the amount of CO_2_ exhausts into the atmosphere is nowadays one of the crucial challenges in the fight against the dramatic climate changes due to global warming. Several recent reviews describe the electrocatalytic and/or photocatalytic reduction of CO_2_ using different molecular catalytic systems [1,2,3,4,5,6,7]. Many of these utilize expensive 4d and 5d transition metals, such as Ru, Rh, Pd, Re, Os, or Ir with bipyridine [8,9,10] or phosphate-like ligands [11,12,13,14]. In addition, mononuclear 3d transition metal (Fe, Ni, Co) complexes with macrocyclic ligands have been reported to be active in the reduction of CO_2_ [4,15,16,17,18,19,20], as well as multinuclear 3d transition metal (Fe, Ni, Cu) complexes [21,22,23,24,25,26], as summarized in several recent reviews [1,2,5,7]. As a further instance, binuclear Cu(II) tetraazacyclotetradecane units show affinity forwards carboxylates, which could be a further interesting option for a CO_2_ reduction strategy [27,28,29,30,31].

It is generally assumed that the affinity of transition metal complexes towards CO_2_, and their ability to reduce it, is closely related to a low oxidation state of the central metal atom, as found in Fe(0)-porphyrins [4], Co(I) and Ni(I) complexes of 1,4,8,11-tetraazacyclotetradecane (denoted as cyclam) [16,17,32,33], or Co(I) and Fe(I) corroles [20,34]. Several cyclam complexes of Co(I,II) [35,36,37,38,39,40,41,42,43] and Ni(I,II) [44,45,46,47,48,49] have already been extensively studied, concerning their potential in electrochemical or photochemical reduction of CO_2_ [1,2,5,7]. Wang et al. [5] summarize the electrocatalytic potentials of CO_2_ reduction with different Ni complexes, the CO_2_ to CO conversion potential of, e.g., [Ni(cyclam)]^2+^ is −1.0 V (H_2_O at pH 4.1 with a Hg working electrode) as found by Sauvage et al. [17], later also confirmed by Froehlich and Kubiak [49] using a glassy carbon electrode (−1.3 V). The lowest potential with respect to CO_2_ to CO conversion pointed out in Wang et al. [5] was achieved with the phosphine dinuclear complex [Ni_2_(l-CNCH_3_)(CNCH_3_)2(l2-dppm)2] (dppm = bis(diphenylphosphino)methane) that could catalyse the electro-reduction of liquid CO_2_ to CO at approximately −0.63 V vs. NHE [21], although a further catalytic reaction was hindered due to the capture of the evolved CO by the dinuclear nickel complex [5]. Despite the extensive experimental research in the field, the insight into their electronic structure and the understanding of the interaction between the central metal atom and the carbon atom of CO_2_ remains relatively limited. The ability of certain macrocyclic complexes to bind CO_2_, in contrast to other quite similar structures, which do not show CO_2_ affinity, has not yet been sufficiently addressed by theoretical methods. Among the few reports published so far, Sakaki [50,51] studied the affinity of Ni(I,II) complexes for CO_2_ with different combinations of NH_3_ and F^-^ ligands. Besides, several studies focused on the frontier orbitals features between the central metal atom and the CO_2_ carbon, to characterize their interaction, including calculations of relevant properties such as electronic transitions [2]. Aside from the oxidation state of the central metal atom, several other experimental studies have considered the impact of π-conjugation of the macrocycle, the presence of an apical co-ligand, and solvent effects on the ability to bind and/or reduce CO_2_ [46,47,48].

Herein, we focus on the analysis of the relation between the electronic structure of 15-membered octaazamacrocyclic nickel complexes and their affinity for CO_2_. We have chosen the simplest nickel(II) complex from a previously reported series (originally denoted [**NiL^1^**], herein abbreviated [**NiL**], see Figure 1a) [52,53], as our trial complex with an almost completely unsaturated (π-conjugated) macrocycle. The conjugation is interrupted by an sp^3^ hybridized methylene group carbon in the 15-membered chelate ring. Interestingly, the hydrogens of the sp^3^-hybridized carbon in [**NiL**] were found to be acidic [52,53]. [**NiL**] does not show any affinity forwards CO_2_ upon reduction (vide infra). In contrast, the hydrogenated (fully saturated) analogue (denoted as [**NiLH**], see Figure 1b), which is similar to the [Ni(cyclam)]^2+^ [2,49,50,54,55,56], is capable of binding CO_2_ upon reduction. The energetics and optimal geometries of the studied [**NiL**] and [**NiLH**] compounds (including the CO_2_ adducts) are compiled in the first two results sections. Afterwards, the Mulliken population analysis (MPA), i.e., the atomic charges and atomic populations, and the quantum theory of atoms in molecules (QTAIM) analysis are utilized to characterize the oxidation state and bonding in the studied complexes. In the last section, theoretical insights into the electronic structure of starting nickel complexes and their CO_2_ adducts is offered from the perspective of localized orbitals, domain averaged fermi holes (DAFH), and frontier orbitals.

This paper is dedicated to prof. Linus Pauling as a part of the Special Issue in his honour. We try to communicate the insights into chemical bonding from the perspective of the different approaches ubiquitous in computational chemistry. Aside from the orbital picture, we explore the common features of distinct theoretical methods and seek the complementarity in diversity, as well as their direct comparison, in the characterization of the bonding interactions in the studied complexes. We particularly focus on the oxidation state of the central Ni atom and the N-C_CO2_ bond.

## 2. Results

### 2.1. Energetics

Total energies and enthalpies of studied compounds ([**NiL**] and [**NiLH**]) are compiled in Appendix A, including the energetics of their optimized geometries with CO_2_ and the free CO_2_ molecule itself. To further distinguish and characterize the complexes studied, the following notation is applied where appropriate: e.g., ^2^[**Ni^I^LH**]^+^ denotes the [**NiLH**] species in a doublet spin state, Ni in oxidation state I and a total charge of +1. The B3LYP/6-311G* reaction energy (*E*) and enthalpy (*H*) were evaluated in the following manner:(1)$$∆E_{R}=E_{NiCO_{2}}-\left(E_{Ni}+E_{CO_{2}}\right)$$
(2)$$∆H_{R}=H_{NiCO_{2}}-\left(H_{Ni}+H_{CO_{2}}\right)$$
(3)$$∆E_{I}=E_{NiCO_{2}}-\left(E_{CO_{2}}^{NiCO_{2}}+E_{Ni}^{NiCO_{2}}\right)$$
where subscript *R* denotes reaction energy/enthalpy and subscript *I* denotes interaction energy. The optimized structure of a given complex is denoted as *Ni*, the complex with CO_2_ is denoted as *NiCO_2_*, and the free carbon dioxide is denoted as *CO_2_* in Equations (1) and (2). In Equation (3), the molecular fragments are denoted as${\;}_{CO_{2}}^{NiCO_{2}}$ and${\;}_{Ni}^{NiCO_{2}}$ and represent the geometry of CO_2_ and the Ni complexes in the particular NiCO_2_ complex, respectively.

The BSSE uncorrected reaction and interaction *E* and *H* values for all the studied species, as well as the counterpoise corrected (CPC) interaction energy $∆E_{I}^{CPC}$ and the distances between Ni and the carbon of CO_2_ (Ni-C_CO2_) are collected in Table 1. According to the B3LYP/6-311G* calculations, the reduced species ^1^[**Ni^0^LH**]^0^ binds CO_2_ in the ground state (see Table 1 and Figure 2d) as well as in the ^2^[**Ni^I^LH**]^+^ state, as found previously for similar aliphatic Ni^I^ complexes [49]. In addition, the reduced species ^3^[**NiL**]^2−^ is also able to bind CO_2_ (see Table 1 and Figure 2c) although the triplet state is above the singlet ground state, hence it has to be considered as an excited state (see Appendix A). The remaining species do not show the ability to bind CO_2_, i.e., the Ni–C_CO2_ distances found are above 3.5 Å and the BSSE uncorrected interaction energies do not exceed −8 kJ mol^−1^. According to the calculated energetics, the ^1^[**Ni^0^LH**]^0^ complex shows a considerably larger affinity to CO_2_ compared to ^3^[**NiL**]^2−^ and ^2^[**NiLH**]^+^, which is true for both ∆*E*_I_ and ∆*E*_R_ (see Table 1), with the CO_2_ molecule being the most bent in ^1^[**Ni^0^LH**]^0^ (see Appendix A). Even though the values obtained with the CPC approach are higher, the conclusions remain the same. Furthermore, it should be noted that when using the CPC approach the solvent effect is not accounted for. For ^2^[**NiLH**]^+^, the interaction energy indicates that CO_2_ binding with this complex is weaker than with ^3^[**NiL**]^2−^ and ^1^[**NiLH**]^0^. The small negative ^2^[**NiLH**]^+^ reaction energy ∆*E*_R_ suggests only a lightly exothermic process, whereas for the other two complexes ^3^[**NiL**]^2−^ and ^1^[**NiLH**]^0^ the binding of CO_2_ is considerably more exothermic (see Table 1). It is pertinent to note that in the calculations of molecular complexes, especially with smaller basis sets as is the case herein, the BSSE plays an important role when deciding about the strength of the interaction. This error can be considerable and lead to overestimated interaction energies (≈108 kJ/mol in the case of ^1^[**Ni^0^LH**]^0^ or 40 kJ/mol in the case of ^2^[**NiLH**]^+^).

Due to the predicted affinity of ^1^[**NiLH**]^0^, ^2^[**NiLH**]^+^, and ^3^[**NiL**]^2−^ for binding CO_2_, these complexes (including their CO_2_ adducts ^1^[**NiLH-CO_2_**]^0^, ^2^[**NiLH-CO_2_**]^+^, and ^3^[**NiL-CO_2_**]^2−^) will be used as the reference compounds in our endeavour to identify the characteristics of the electronic structure that promotes a high affinity forward CO_2_. This analysis will be further extended by comparison with chosen [**NiL**] complexes to examine in further detail their failure to bind CO_2_, namely ^1^[**NiL-CO_2_**]^0^ and ^1^[**NiL_fr_CO_2_**]^0^. The complex ^1^[**NiL-CO_2_**]^0^ represents the optimized geometry, with d(Ni–C_CO2_) being equal to 3.576 Å, i.e., no Ni–C_CO2_ bond (see Table 1). For the geometry optimization of ^1^[**NiL-_fr_CO_2_**]^0^ the d(Ni-C_CO2_) distance has been frozen at 1.913 Å as found in ^1^[**NiLH-CO_2_**]^0^, so as to simulate an artificially constrained system and to observe the resulting effect on the electronic structure.

### 2.2. Optimized Geometries

Bond distances and bond angles of chosen B3LYP/6-311G* optimized complexes are summarized in Appendix A. The studied [**NiL**] complexes contain a 15-membered octaazamacrocyclic ligand with one sp^3^-hybridized carbon atom in the macrocycle, which leads to a non-planar geometry (see Figure 1). Optimized structures of all species derived from the [**NiL**] complexes show only a small deviation from planarity in the NiN_4_ coordination polyhedron, angles NiN(1)N(2) and NiN(5)N(6) are between 132–133° and 131–132°, respectively. However, a significant distortion from the planar geometry can be seen when saturating the macrocyclic ligand with hydrogen atoms (i.e., for complex ^1^[**NiLH**]^2+^, see Figure 1) which also leads to the out of the plane displacement of the N(2) atom (NiN(1)N(2) = 110.77°). In general, when comparing [**NiLH**] to the [**NiL**] complexes, a slight elongation of the N-N bond lengths in [**NiLH**] can be observed, as well as further bending within the coordination polyhedron (see Appendix A). Furthermore, in the case of the ^1^[**NiLH**]^0^ complex, we observe that the Ni–N(1) and Ni–N(5) distances become grossly elongated in comparison to those in ^1^[**NiLH**]^2+^. Hence, the coordination polyhedron in ^1^[**NiLH**]^0^ is considerably distorted from a square-planar geometry.

Changes in the geometry upon CO_2_ binding are the most pronounced for the ^1^[**NiLH**]^0^ complex. The shortening of the and Ni–N(5) (−0.5 Å) distance shows the largest difference, while the Ni–N(1) becomes 3.08 Å long, see Appendix A. The CO_2_ molecule itself is bent in the complex, leading to the O–C–O angle of 129° in ^1^[**NiLH-CO_2_**]^0^, 147° in ^2^[**NiLH-CO_2_**]^+^, and 138° in ^3^[**NiL-CO_2_**]^2−^, in contrast to the linear geometry (180°) of the gas phase CO_2_ molecule, or CO_2_ that did not directly bind to the Ni centre in the ^1^[**NiL-CO_2_**]^0^ complex (see Figure 2a and Appendix A). In ^1^[**NiL_fr_CO_2_**]^0^ (see Figure 2b), the structure optimization also resulted in a bent CO_2_ geometry (145°).

### 2.3. MPA and QTAIM Charges

MPA Ni charges (see Table 2 and Appendix A) agree well with those obtained from QTAIM (see Table 2 and Appendix A). The MPA charges of the remaining atoms in the studied systems are compiled in Appendix A, and the QTAIM charges are shown in Appendix A. The difference between MPA and QTAIM charges are particularly large for the carbon and oxygen atoms of CO_2_.

According to MPA, the Ni charges of [**NiL**] are close to one (Table 2) in all of the investigated oxidation states of these complexes. This points to a ligand-centred reduction for both the one and two-electron reduced species, without a significant change on the central atom. For the two-electron reduced ^1^[**NiL**]^2−^ species, the Ni MPA charge changes only by 0.11 e compared to ^1^[**NiL**]^0^ and by 0.27 e for ^3^[**NiL**]^2−^. Still, the AO populations in the case of ^1^[**NiL**]^2−^ change only due to a stronger Ni ligand interaction (raise of the s_σ_ and $d_{x^{2}-y^{2}}$ populations), while the presence of $d_{x^{2}-y^{2}}$ spin density suggests a reduction of Ni to the oxidation state I in ^3^[**NiL**]^2−^, see Table 3 and the coming section. In the [**NiLH**] complex, we observed a decrease in the Ni charge upon subsequent reductions from one to almost zero (1.05 e for ^1^[**NiLH**]^2+^, 0.76 e for ^2^[**NiLH**]^+^ and 0.04 e for ^1^[**NiLH**]^0^), which points to the reduction on the central atom, unlike in the ^2^[**NiL**]^−^ and ^1^[**NiL**]^2−^ complexes. These physical MPA charges can be attributed to oxidation states II, I, and 0 in the case of ^1^[**NiL**]^0^/^1^[**NiLH**]^2+^, ^3^[**NiL**]^2−^/^2^[**NiLH**]^+^ and ^1^[**NiLH**]^0^, respectively. More details on the considered oxidation states will be given below (see the MPA d-orbitals Ni populations, Localized orbitals and DAFH analysis sections).

The presented charges point to an increase of the Ni charge when the CO_2_ molecule binds to ^1^[**NiLH**]^0^, ^2^[**NiLH**]^+^, and ^3^[**NiL**]^2−^ (see Table 2 and Appendix A), i.e., a charge transfer from Ni to CO_2_ is observed, where Ni is oxidized and CO_2_ is reduced. The most significant change in Ni charge in the presence of CO_2_ is found for ^1^[**NiLH-CO_2_**]^0^, with a loss of ca. 0.8 (0.53) e when compared to ^1^[**NiLH**]^0^ MPA (QTAIM) charges. Furthermore, the bound CO_2_ molecule in ^1^[**NiLH-CO_2_**]^0^ has an MPA (QTAIM) charge of −0.93 (−0.98) e. Hence, the central Ni atom becomes oxidized (from 0 to I) and the CO_2_ molecule reduced (formally from 0 to −1) when comparing ^1^[**NiLH-CO_2_**]^0^ with ^1^[**NiLH**]^0^ and the free CO_2_ molecule (see Table 2). In ^2^[**NiLH-CO_2_**]^+^ and ^3^[**NiL-CO_2_**]^2−^, the Ni MPA (QTAIM) charge has increased by ca. +0.39 and +0.38 (+0.23 and +0.20) e with respect to ^2^[**NiLH**]^+^ and ^3^[**NiL**]^2−^. The bound CO_2_ molecule carries an MPA (QTAIM) charge of −0.47 (−0.46) e in ^2^[**NiLH-CO_2_**]^+^ and −0.71 (−0.66) e in ^3^[**NiL-CO_2_**]^2−^. On the contrary, an Ni charge of ^1^[**NiL_fr_CO_2_**]^0^ and ^1^[**NiL-CO_2_**]^0^ changes only slightly or remains unchanged, when compared to ^1^[**NiL**]^0^. Hence, the polarization of the CO_2_ molecule experiences only a slight change in ^1^[**NiL-CO_2_**]^0^ when compared to the free CO_2_ (see Table 2). In ^1^[**NiL_fr_CO_2_**]^0^, the CO_2_ molecule gains 0.43 (0.39) e based on MPA (QTAIM), although the actual change of Ni charge is only 0.1 e. Nevertheless, such charge transfer, from the macrocycle ligand to CO_2_, seems to be energetically disfavoured.

### 2.4. MPA d-Orbital Ni Populations

The B3LYP/6-311G* MPA d-orbital populations, as well as the excess of s_σ_ (sum of s-population minus six electrons assumed to occupy 1s, 2s and 3s shells) of the central Ni atom, are summarized in Table 3. Eight closed-shell d electrons (d_xz_, d_yz,_ d_xy,_
$d_{x^{2}-y^{2}}$ and $d_{z^{2}}$) have been found on Ni for all [**NiL**] species, as well as in the case of the ^1^[**NiLH**]^2+^ complex, and therefore one can assign a formal oxidation state of II to the nickel (albeit the physical, and/or total, d population is between 8.3–8.5). In the case of ^1^[**NiLH**]^0^, the value of the total d population (d_total_) is close to nine (8.8 e) which together with the 1.1 e s_σ_ population indicate an oxidation state of 0 on nickel in ^1^[**NiLH**]^0^. In particular, the occupation of Ni d-orbitals agrees with the crystal field theory prediction for a square-planar geometry of both [**Ni^II^L**] and [**Ni^II^LH**]-like species, with the doubly occupied d_xz_, d_yz_ and d_xy_ orbitals, an almost doubly occupied $d_{z^{2}}$, and the $d_{x^{2}-y^{2}}$ orbital, involved in the σ coordination. When comparing ^1^[**NiL**]^0^ with the oxidized (^2^[**NiL**]^+^ and ^1^[**NiL**]^2+^) and reduced (^2^[**NiL**]^−^) species, there are no significant changes in the d-orbital populations. On the other hand, comparing ^1^[**NiL**]^0^ with ^3^[**NiL**]^2−^, one can identify an increase in the $d_{x^{2}-y^{2}}$ (0.35 e) and s_σ_ (0.1 e) populations for the doubly charged species ^3^[**NiL**]^2−^ (including the presence of a nonnegligible spin population in $d_{x^{2}-y^{2}}$, see Table 3). In the case of [**NiLH**], the most significant changes have been revealed for the ^1^[**NiLH**]^0^ complex, where the $d_{z^{2}}$ population decreased by 0.09 e and $d_{x^{2}-y^{2}}$ and s_σ_ increased their populations by 0.35 and 0.48 e, respectively, when compared to ^1^[**NiLH**]^2+^. In ^2^[**NiLH**]^+^, a significant change of −0.36 e is observed for the $d_{x^{2}-y^{2}}$ orbital population compared to ^1^[**NiLH**]^2+^. Thus, the individual d-populations closely follow the conclusion derived from the MPA and QTAIM charge analysis that the reduction of [**NiLH**] occurs on the central atom unlike in [**NiL**] complexes. Still, the particular d-populations of ^1^[**NiLH**]^0^ can be less straightforwardly assigned to a close-shell or formally open-shell character as suggested in the crystal field theory. This is caused by the large distortion of the coordination polyhedron from a square-planar structure in ^1^[**NiLH**]^0^ as mentioned above.

For completeness, Ni atomic orbitals that change their populations when the studied complexes bind to CO_2_ are $d_{x^{2}-y^{2}}$, $d_{z^{2}}$, and s_σ_. This is especially true for ^1^[**NiLH-CO_2_**]^0^, where $d_{z^{2}}$ and s_σ_ decrease their populations by 0.32 and 0.57 e, respectively, and $d_{x^{2}-y^{2}}$ increases by 0.11 e. In the case of ^3^[**NiL-CO_2_**]^2−^ and ^1^[**NiL_fr_CO_2_**]^0^, $d_{z^{2}}$ populations decrease by ca. 0.3 e, s_σ_ a decrease of 0.1 e while $d_{x^{2}-y^{2}}$ and d_xy_ becomes only slightly more populated. A similar situation is also found in the ^2^[**NiL-CO_2_**]^+^, where $d_{z^{2}}$ and s_σ_ populations decrease by ca. 0.24 e, while $d_{x^{2}-y^{2}}$ remains unchanged.

### 2.5. QTAIM BCP Analysis

Additional information on the electronic structure, besides the robust atomic charges (spins) discussed above, can be obtained from the QTAIM analysis including the bond critical point (BCP) characteristics (electron density (ρ), its Laplacian (∆ρ) and ellipticity (ε)) and the delocalization index (DI). DI can be used to estimate the formal bond order and/or bond strength. BCP characteristics and DIs are compiled in Appendix A. The change in the Ni–N BCP characteristics upon reduction are the most obvious for ^1^[**NiLH**]^0^. The Ni–N(1) and Ni–N(5) bond lengths become elongated after reduction to 2.96 and 2.77 Å (BCPs are still present), which leads to lower values of BCP ρ, ∆ρ, and DIs and we observed a significant increase in the value of Ni–N(1) and Ni–N(5) BCP ellipticities ε= 0.79 and 0.38, respectively. The Ni–N(5) bond becomes shorter once CO_2_ binds to the complex and this BCP ellipticity value in ^1^[**NiLH-CO_2_**]^0^ decreases and becomes numerically reasonable, see in the coming text. The problem with the Ni–N(1) and Ni–N(5) ellipticities of ^1^[**NiLH**]^0^ is the large interatomic distance which leads to small BCP Hessian eigenvalues (λ_i_), which grossly affect the ellipticity (where ellipticity is defined as ε = λ_1_/λ_2_ – 1), BCP Hessian eigenvalues are presented in Appendix A. Although this issue does not completely diminish for Ni–N(1) with a bond length of 2.97 Å and ε = 0.33. The Ni–N(3) and Ni–N(7) bonds become slightly longer when comparing ^1^[**NiLH-CO_2_**]^0^ and ^1^[**NiLH**]^0^, which similarly affects the particular BCP characteristics, as found for ^3^[**NiL**]^2−^. In the ^1^[**NiLH-CO_2_**]^0^ complex, the BCP value of ρ (0.93), ∆ρ (3.50), as well as the value of DI (0.86) of the Ni–C_CO2_ bond are the largest, indicating the strongest dative interaction between the [**NiLH**] complex and CO_2_, which naturally correlates with the shortest Ni–C_CO2_ bond length. In the ^3^[**NiL-CO_2_**]^2−^ and ^2^[**NiLH-CO_2_**]^+^ complexes, the changes in the Ni–N BCP characteristics after binding of the CO_2_ molecule are not dramatic, and we observe only a small elongation of the Ni–N bonds, which in turn leads to lower values of BCP ρ, ∆ρ, and DIs. When compared to the free CO_2_, we have also observed a small increase of the C–O bond ellipticity which correlates with the bent structure of the bound CO_2_. Still, the C–O DIs even in ^1^[**NiLH-CO_2_**]^0^ decrease only by 0.12 with respect to the free CO_2_. In the case of ^1^[**NiL_fr_CO_2_**]^0^ with the Ni–C_CO2_ distance frozen, as well as in the case of ^2^[**NiLH-CO_2_**]^+^, we have not observed any differences in the C–O BCP characteristics compared to the free CO_2_, despite a non-zero Ni–C_CO2_ DI. In the case of ^1^[**NiL-CO_2_**]^0^, the Ni–C_CO2_ BCP has very low ρ, ∆ρ, and DI values (0.003, 0.010 and 0.008, respectively) and the outcome of the QTAIM analysis when comparing the **NiL** unit in ^1^[**NiL-CO_2_**]^0^ and ^1^[**NiL**]^0^ is the same (not shown).

### 2.6. Localized Orbitals, Frontier Orbitals and DAFH Analysis

To further elucidate the relation between the electronic structure and the CO_2_ affinity of the studied complexes, localized orbitals (LOC) and domain averaged Fermi holes (DAFH) analysis has been performed. In the present cases, the Ni and CO_2_ atoms have been chosen as DAFH domains for the inspection of the Ni-C_CO2_ bonding situation. As mentioned in the Methods—Computation Details section, the LOC and DAFH eigenvectors were analysed using the Ni-C_CO2_ bond aligned geometries (the Ni-C_CO2_ bond was aligned in the *z*-axis direction). LOC populations are shown in Table 4 and DAFH eigenvectors and eigenvalues are compiled in Figure 3. The essential Ni-C_CO2_ bond contributions are coming from the $d_{z^{2}}$ atomic orbital of the Ni atom and in the C atoms case, from 2s and 2p_z_ atomic orbitals (AOs), according to LOC populations in Table 4 and the DAFH eigenvector shapes in Figure 3. The localized d-orbitals on the central atom are compiled in Appendix A, indicating a d^8^ electronic configuration in the [**NiL**] complexes except for ^3^[**NiL**]^2−^. In the ^3^[**NiL**]^2−^ case, the localization procedure yields a formal d population of 9 on the Ni atom, which is distributed over 5α and 4β LOCs. In the ^1^[**NiLH**]^0^, ^2^[**NiLH**]^+^ and ^1^[**NiLH**]^2+^ complexes, the localization procedure leads to formally d^10^, d^9^, and d^8^ Ni LOCs, which correspond to the formal oxidation states 0, I, and II, respectively. The DAFH analysis of ^1^[**NiLH**]^0^ and ^1^[**NiL**]^0^ agrees with the d^10^ and d^8^ configuration, 5 and 4 eigenvectors with occupation numbers [1.93, 1.92, 1.88, 1.83, 1.62] and [1.92, 1.92, 1.87, 1.85] are present in the Ni basin, respectively. In the ^1^[**NiLH-CO_2_**]^0^, the DAFH eigenvectors indicate a small polarization of the Ni–C_CO2_ bond towards the CO_2_ molecule, i.e., the Ni basin contains 1.04 electrons, as well as the CO_2_ basin, in the Ni–C_CO2_ eigenvector/bond (see Figure 3). Both DAFH eigenvectors show an interaction between the 2s and 2p_z_ orbitals of the carbon atom and the $d_{z^{2}}$ orbital of nickel. The DAFH analysis is in reasonable agreement with the results of the LOC procedure. Nevertheless, the LOC procedure shows a larger polarization of the Ni–C_CO2_ bond with 1.48 electrons on the C atom and 0.80 electrons on the Ni atom (see Table 4), which would indicate a more dominant contribution from C to the unevenly shared electron pair and the σ(Ni–C_CO2_) bond having a Ni(δ+)-CO_2_(δ−) nature. (This overestimated polarization seems to be an artefact of the AO based analysis in the LOC procedure when compared to the QTAIM based atoms definition in DAFH). In the ^2^[**NiLH-CO_2_**]^+^ and ^3^[**NiL-CO_2_**]^2−^, the polarization of the Ni-C_CO2_ bond is opposite to ^1^[**NiLH-CO_2_**]^0^ and these Ni-C_CO2_ DAFH eigenvectors show the dominant contribution from the Ni atom. More specifically, the Ni basin contains the following part of the Ni-C_CO2_ eigenvectors: α(0.88)/β(0.63) for ^2^[**NiLH-CO_2_**]^+^ and α(0.84)/β(0.57) for ^3^[**NiL-CO_2_**]^2−^(a total two electrons). As a complement, the CO_2_ basin has the following composition of eigenvector shared with Ni: α(0.13)/β(0.38) and α(0.21)/β(0.47), respectively, see Figure 3. The LOC procedure is in qualitative agreement with the particular DAFHs of ^2^[**NiLH-CO_2_**]^+^ (see Table 4 and Figure 3). In ^1^[**NiL_fr_CO_2_**]^0^, the contribution of the C atom to the unevenly shared electron pair of this bond is only 0.44 electrons for both methods (see Table 4 and Figure 3). However, it should be noted that this is an enforced interaction between Ni and C.

To illustrate further differences between ^1^[**NiL**]^0^ and ^1^[**NiLH**]^0^, we will briefly discuss the frontier orbitals, see Appendix A (pictorial representation without CO_2_) as well as S3a and S3b (pictorial representation with CO_2_). From Appendix A, it can be seen that the highest occupied molecular orbital (HOMO) as well as the lowest unoccupied molecular orbital (LUMO) of ^1^[**NiL**]^0^ are both localized mainly on the macrocycle ligand, and are involved in π interactions with only a small antibonding contribution of d(Ni) orbitals. The HOMO-1 of ^1^[**NiL**]^0^ has a 24.7% d_yz_ contribution, see Appendix A. Additional d contributions of Ni in MOs of ^1^[**NiL**]^0^ can be found in the lower orbitals HOMO-4 (70.7% $d_{z^{2}}$), HOMO-6 (34.9% d_xz_), HOMO-7 (29.0% d_xz_), or HOMO-10 (53.7% d_yz_) (not shown). The situation with the Ni d-character in the frontier orbitals of ^1^[**NiL**]^2−^ is not much different from ^1^[**NiL**]^0^. The HOMO and HOMO-1 orbitals have π ligand character and the Ni d-population becomes significant in the lower orbitals, i.e., HOMO-2 (31.7% d_xz_), HOMO-3 (30.4% $d_{z^{2}}$, 17.9% d_yz_), HOMO-4 (46.5% $d_{z^{2}}$, 14.1% d_yz_), HOMO-5 (42.7% d_yz_), or HOMO-9 (35.9% d_xz_ and 11.2% d_xy_) (not shown). On the other hand, in the ^1^[**NiLH**]^0^ complex, HOMO to HOMO-4 are all localized predominantly on Ni (see HOMO and HOMO-1 in Appendix A). In ^3^[**NiL**]^2−^, HOMO-1, HOMO, and LUMO have a d(Ni) character, although being still involved in antibonding or non-boning d(Ni)-p(N) interactions with the macrocycle ligand (see Appendix A). The Ni-C_CO2_ interactions in ^1^[**NiLH-CO_2_**]^0^, ^2^[**NiLH-CO_2_**]^+^, and ^3^[**NiL-CO_2_**]^2−^ can be seen in the plots of HOMO, β-HOMO, and β-LUMO, respectively, see Appendix A. It is still fair to stress that for example in ^1^[**NiLH-CO_2_**]^0^, it is not only the HOMO that is involved in the Ni-CO_2_ interactions, but also, e.g., HOMO-1 to HOMO-4. Hence, the bonding picture taken with respect to a single MO can have some qualitative significance, but additional bonding features are often spread among a larger number of MOs due to the orthonormality conditions of the Hartree–Fock (Kohn-Sham) equations.

For instance, the HOMO of ^1^[**NiLH-CO_2_**]^0^ has a 67.4% Ni and 22.3% CO_2_ composition (MPA%), which formally gives a bond description like that present in ^1^[**NiL_fr_CO_2_**]^0^ LOCs in Table 4, i.e., the bond polarization is not appropriate, suggesting a Ni to CO_2_ dative (donor) interaction. If we consider ^3^[**NiL-CO_2_**]^2−^, the HOMO and HOMO-1 do not show a bonding contribution at all, although CO_2_ is bound. Still, the frontier orbitals are useful when identifying the locus of a redox process (addition or removal of an electron), or when arguing about how the ligand π-system in [**NiL**] hampers the metal reduction, which is not the case in [**NiLH**].

## 3. Methods-Computational Details

Geometry optimizations of [**NiL**] and [**NiLH**] complexes with different charges and/or in various spin states were performed using the Gaussian16 [57] program suite employing the B3LYP/6311-G* computational protocol [58,59,60,61,62,63,64]. The unrestricted DFT formalism was used for open-shell systems (UB3LYP). The effect of dichloromethane as a solvent was approximated via the integral equation formalism polarizable continuum model (IEFPCM) [65,66] as implemented in Gausian16. The stability of the optimized structures was confirmed by vibrational analysis. The counterpoise correction (CPC) method was applied to the interaction energy calculations to mitigate the basis set superposition error (BSSE), without including solvent effect [67,68].

The Ni–C_CO2_ bond character was analysed using the domain-averaged fermi holes (DAFHs) [69,70,71] in the DGrid 5.1 program package [72] employing the B3LYP/6-311G* fchk files from the Gaussian16 program package. The localized orbitals (LOC) and their Mulliken atomic population analysis (MPA) were obtained via the ORCA 4.2.0. package [73]. The cartesian z-axis has been aligned with the particular Ni–C_CO2_ bond (denoted along bond aligned geometries), to allow for a further analysis of the AO contributions. The exploration of the topology of electron density utilized the quantum theory of atoms in molecules (QTAIM) [74] analysis using the AIMAll [75] package and the fchk Gaussian16 files. The Molekel package [76] was used for the visualization of the molecular orbitals and DAFH eigenvectors.

## 4. Summary and Outlook

The electronic structure of the ^1^[**NiL**]^0^ complex is compared with the hydrogenated ^1^[**NiLH**]^2+^ analogue. The analysis of the single and two-electron reduction has been performed and affinity of CO_2_ binding has been explored. Our results indicate that the CO_2_ reduction could possibly be achieved by involving the investigated complexes as mediators in electrocatalysis or electro-photocatalysis. We have identified a change of the oxidation state of the central Ni atom upon the reduction of ^1^[**Ni^II^LH**]^2+^, which leads to the affinity of Ni^I^/Ni^0^ in the reduced species to CO_2_. Subsequently, CO_2_ is (partially) reduced and the creation of the Ni^II^-CO_2_^δ−^/Ni^I^-CO_2_^−^ couple adduct as described in the literature [1,2] is confirmed. We have identified only a single Ni–C_CO2_ σ-bond even in the doubly reduced ^1^[**NiLH-CO_2_**]^0^ species. In the case of [**NiL**], the one or two-electron reduction does not change the oxidation state of Ni^II^. Hence, no affinity of the central atom in [**NiL**] towards CO_2_ is found in the ground state, while the triplet (excited) state of the double reduced ^3^[**NiL**]^2−^ species is capable of binding CO_2_.

The DAFH eigenvectors, as well as LOC orbitals, indicate a Ni–C_CO2_ polarization from the reduced CO_2_^−^ species towards Ni^I^ in ^1^[**NiLH-CO_2_**]^0^, with the Ni–C_CO2_ bond being the strongest among the studied CO_2_ complexes (according to DI, energetics and/or the bond length). In ^1^[**NiL_fr_CO_2_**]^0^, one finds an enforced Ni($d_{z^{2}}$) to C coordination (dative interaction). The Ni–C_CO2_ interaction in ^2^[**NiLH-CO_2_**]^+^ and ^3^[**NiL-CO_2_**]^2−^can be considered an intermediate between ^1^[**NiLH-CO_2_**]^0^ in the β DAFHs or LOCs and ^1^[**NiL_fr_CO_2_**]^0^ in the α ones.

Hence, the activation of [**Ni^II^L**] towards CO_2_ binding would require a two-electron reduction and an optical promotion to the triplet state to activate the Ni^I^ oxidation state. Otherwise, the π-conjugated (unsaturated) ligand remains the locus of the [**Ni^II^L**] reduction and the Ni^II^ central atom shows no affinity forwards binding or reducing CO_2_. In the case of ^1^[**Ni^II^LH**]^2+^, the single electron reduction is already sufficient to reduce the central Ni^II^ atom to Ni^I^ and thus activate the affinity for bind and (partially) reduce CO_2_. A further reduction step of ^2^[**Ni^I^LH**]^+^ leads to a considerably distorted geometry from the square planar one, with a Ni^0^ central atom, but the strong affinity forwards binding and reducing CO_2_ leads to a smaller distortion in ^1^[**NiLH-CO_2_**]^0^ when compared to ^1^[**Ni^0^LH**]^0^. Hence, after the first reduction step, [**Ni^I^LH**] binds CO_2_ and is also able to accept one more electron with an electron transfer to the CO_2_ moiety and forming a stable ^1^[**NiLH-CO_2_**]^0^ species, with the CO_2_ moiety experiencing the largest bend among the studied species. A further challenge to the possible working catalytic cycle of ^1^[**NiLH-CO_2_**]^0^ is the split into [**Ni^I^LH**]^+^ and CO_2_^−^ species and/or to avoid the ^1^[**NiLH**]^0^ species occurrence, which can be assumed to be unstable due to the considerable distortion of the coordination polyhedron.

## Figures and Tables

**Figure 1 molecules-26-04139-f001:**
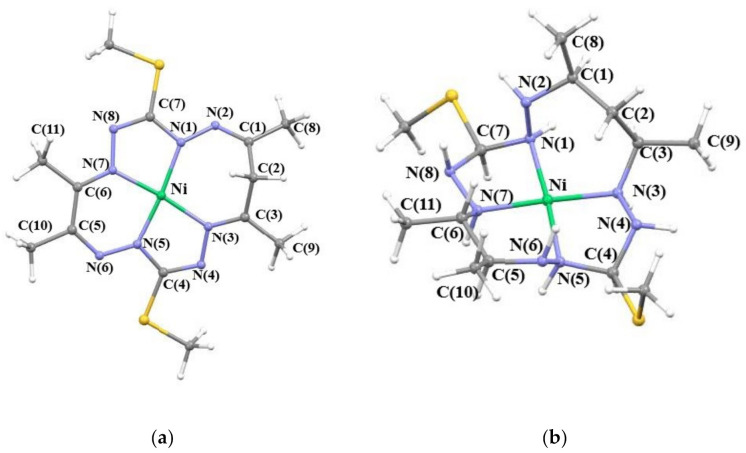
Optimized molecular structures of (**a**) ^1^[**NiL**]^0^ and (**b**) ^1^[**NiLH**]^2+^ complexes with atom labelling.

**Figure 2 molecules-26-04139-f002:**
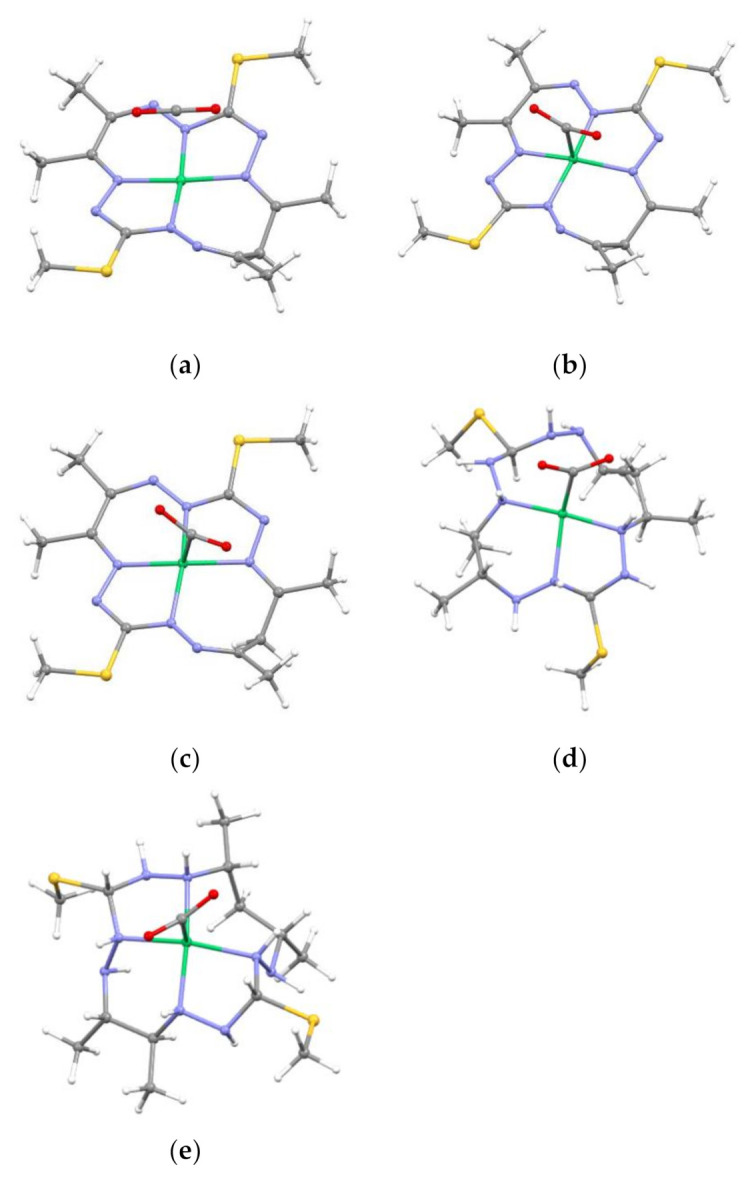
Structure of studied complexes with CO_2_: (**a**) ^1^[**NiL-CO_2_**]^0^, (**b**) ^1^[**NiL_fr_CO_2_**]^0^, (**c**) ^3^[**NiL-CO_2_**]^2−^, (**d**) ^1^[**NiLH-CO_2_**]^0^, and (**e**) ^2^[**NiLH-CO_2_**]^+^. Colour scheme: green Ni, yellow S, blue N, grey C, and white H.

**Figure 3 molecules-26-04139-f003:**
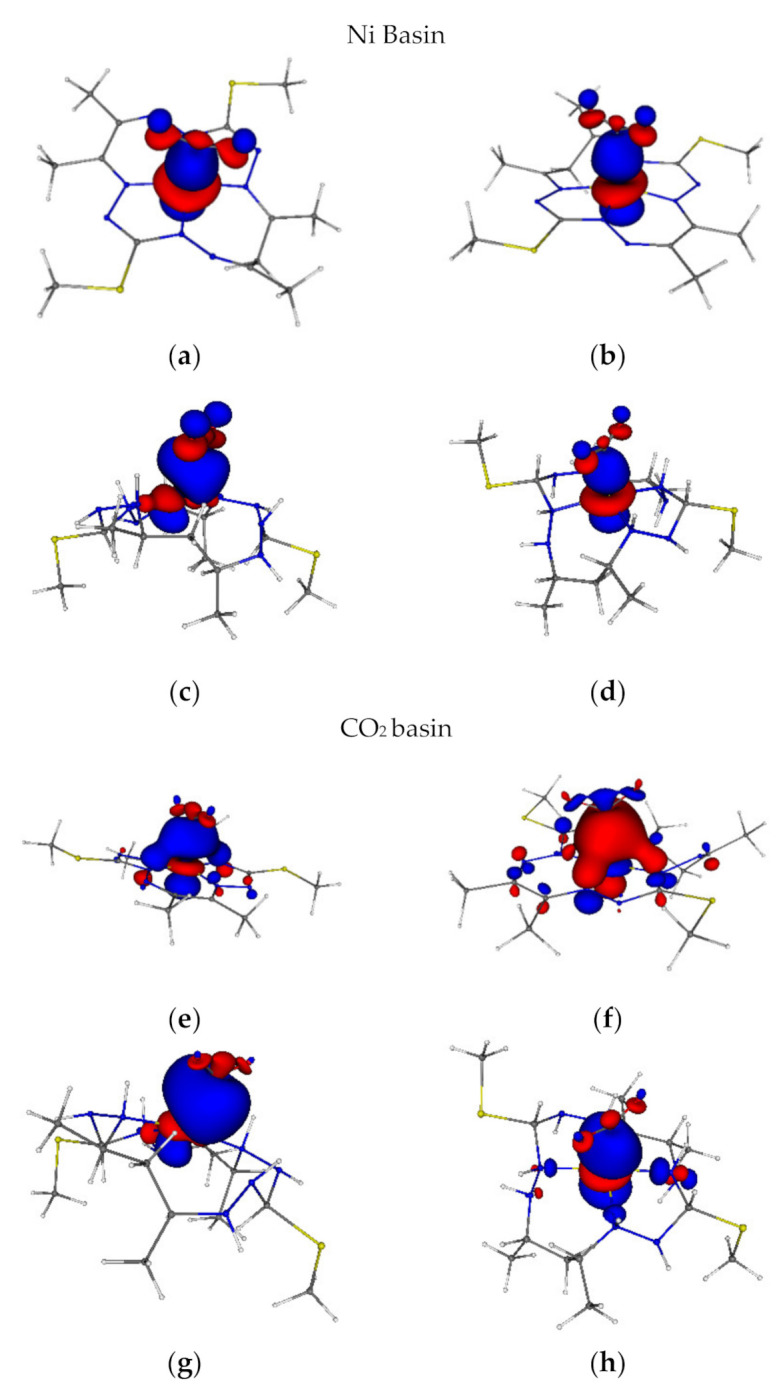
(**a**) ^1^[**NiL_fr_CO_2_**]^0^ (1.62); (**b**) ^3^[**NiL-CO_2_**]^2-^ α(0.84)/β(0.57); (**c**) ^1^[**NiLH-CO_2_**]^0^ (1.04); (**d**) ^2^[**NiLH-CO_2_**]^+^ α(0.88)/β(0.63); (**e**) ^1^[**NiL_fr_CO_2_**]^0^ (0.44); (**f**) ^3^[**NiL-CO_2_**]^2-^ α(0.21)/β(0.47); (**g**) ^1^[**NiLH-CO_2_**]^0^ (1.04); (**h**) [**NiLH-CO_2_**]^+^ α(0.13)/β(0.38). DAFH Ni-C_CO2_ eigenvectors at B3LYP/6-311G* level of theory (isovalue 0.04, red is negative and blue is positive). The Ni atom and CO_2_ have been chosen as the domain to analyse the bonding situation via DAFH analysis. DAFH eigenvalues for the particular basin/domain are given in parentheses.

**Table 1 molecules-26-04139-t001:** The reaction and interaction energies/enthalpies of the studied [**NiL**] and [**NiLH**] complexes with CO_2_ in different spin states, including Ni–C_CO2_ bond lengths.

	∆*E*_R_ [kJ mol^−^^1^]	∆*E*_I_ [kJ mol^−1^]	∆*E*_I_^CPC^ [kJ mol^−1^]	∆*H*_R_ [kJ mol^−1^]	d(Ni-C_CO2_) [Å]
^3^[**NiL**]^2−^	−27.16	−210.59	−159.49	−88.13	2.051
^1^[**NiL**]^2−^	−7.70			−1.28	3.689
^2^[**NiL**]^−^	−5.27			−1.55	3.511
^1^[**NiL**]^0^	−4.37	−11.08	1.21	−0.33	3.576
^2^[**NiL**]^+^	−2.68			−1.11	3.603
^1^[**NiL**]^2+^	−1.86			2.83	3.629
^1^[**NiLH**]^0^	−160.655	−451.90	−343.55	−150.26	1.887
^2^[**NiLH**]^+^	−7.906	−106.86	−66.32	−3.24	2.079

More negative values imply a more favoured bonding interaction.

**Table 2 molecules-26-04139-t002:** B3LYP/6-311G* MPA and QTAIM charges of chosen atoms of the studied complexes.

	Ni		C		O *	
	MPA	QTAIM	MPA	QTAIM	MPA	QTAIM
^3^[NiL]^2−^	0.73	0.74	-	-	-	-
^1^[NiL]^2−^	0.89	0.89	-	-	-	-
^2^[NiL]^-^	0.96	0.94	-	-	-	-
^1^[NiL]^0^	1.00	0.96	-	-	-	-
^2^[NiL]^+^	1.05	0.99	-	-	-	-
^1^[NiL]^2+^	1.10	1.03	-	-	-	-
^3^[NiL-CO_2_]^2−^	1.06	0.94	0.11	1.66	−0.41	−1.16
^1^[NiL_fr_CO_2_]^0^	1.10	1.00	0.21	1.81	−0.32	−1.10
^1^[NiL-CO_2_]^0^	1.00	0.96	0.48	2.15	−0.25	−1.08
^1^[NiLH]^0^	0.04	0.15	-	-	-	-
^2^[NiLH]^+^	0.76	0.66	-	-	-	-
^1^[NiLH]^2+^	1.05	0.94	-	-	-	-
^1^[NiLH-CO_2_]^0^	0.84	0.68	0.03	1.44	−0.48	−1.21
^2^[NiLH-CO_2_]^+^	1.15	0.89	0.19	−0.10	−0.33	−0.08
CO_2_	-	-	0.53	2.15	−0.26	−1.08

* average value of the O atoms charge from the CO_2_ molecule

**Table 3 molecules-26-04139-t003:** B3LYP/6-311G* formal oxidation state of Ni and Mulliken d-orbital populations on the central atoms of the studied complexes (spin population for chosen triplet state complexes in parenthesis).

	Formal ox. State of Ni	$d_{z^{2}}$	d_xz_	d_yz_	$d_{x^{2}-y^{2}}$	d_xy_	s_σ_	d_total_
^3^[NiL]^2−^	I	1.745	1.878	1.830	1.146	1.886	0.693	8.487
		(0.043)	(0.009)	(0.032)	(0.768)	(0.006)	(−0.055)	(0.858)
^1^[NiL]^2−^	II	1.797	1.925	1.903	0.846	1.910	0.684	8.381
^2^[NiL]^-^	II	1.820	1.924	1.908	0.796	1.912	0.639	8.360
^1^[NiL]^0^	II	1.835	1.920	1.906	0.791	1.911	0.599	8.363
^2^[NiL]^+^	II	1.851	1.903	1.908	0.790	1.913	0.562	8.365
^1^[NiL]^2+^	II	1.868	1.893	1.886	0.803	1.915	0.527	8.365
^3^[NiL-CO_2_]^2−^	II	1.420	1.918	1.915	1.205	1.911	0.553	8.343
		(0.273)	(0.009)	(0.012)	(0.774)	(0.009)	(−0.012)	(1.077)
^1^[NiL_fr_CO_2_]^0^	II	1.562	1.922	1.938	0.951	1.923	0.490	8.296
^1^[NiL-CO_2_]^0^	II	1.833	1.919	1.907	0.794	1.911	0.602	8.364
^1^[NiLH]^0^	0	1.649	1.861	1.890	1.527	1.829	1.135	8.756
^2^[NiLH]^+^	I	1.736	1.919	1.908	1.123	1.897	0.662	8.583
^1^[NiLH]^2+^	II	1.836	1.941	1.933	0.762	1.928	0.588	8.400
^1^[NiLH-CO_2_]^0^	I	1.329	1.892	1.870	1.636	1.890	0.577	8.617
^2^[NiLH-CO_2_]^+^	II	1.504	1.928	1.929	1.165	1.923	0.426	8.448

**Table 4 molecules-26-04139-t004:** B3LYP/6-311G* LOC AO populations and atomic total contributions.

	AO Populations				Total	
	s(Ni)	$d_{z^{2}}(\mathrm{Ni})$	s(C)	p_z_(C)	Ni	C
^1^[NiL_fr_CO_2_]^0^	0.18	1.38	0.17	0.27	1.56	0.44
^3^[NiL-CO_2_]^2−^ α	0.08	0.74	0.04	0.07	0.82	0.11
^3^[NiL-CO_2_]^2−^ β	0.12	0.36	0.32	0.30	0.48	0.62
^2^[NiLH-CO_2_]^+^ α	0.09	0.81	0.02	0.05	0.93	0.07
^2^[Ni LH-CO_2_]^+^ β	0.14	0.53	0.17	0.23	0.66	0.40
^1^[Ni LH-CO_2_]^0^	0.18	0.62	0.62	0.52	0.80	1.16

## Data Availability

Not applicable.

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
