# Peer review of "Ni Oxidation State and Ligand Saturation Impact on the Capability of Octaazamacrocyclic Complexes to Bind and Reduce CO2"

_molecules, 2021, doi:10.3390/molecules26144139_

Round 1

Reviewer 1 Report

In this manuscript, Venosova et al. present an extended theoretical study of CO2 binding by Ni macrocyclic complexes (octaazamacrocycles). A sety of theoretical methods was used, and results obtained by each agree well with others.

I could not detect any flaws. In my opinion, the work is suitable for publication in its present form.

The only recommendation (which is, however, not obligatory) belongs to references. There were relevant papers on Cu complexes with similar ligands:

a) 10.2298/jsc0109597m

b) 10.2298/JSC190710088D

c) 10.1016/j.molstruc.2018.10.027

d) 10.2298/JSC140404044T

e) 10.1016/j.molstruc.2013.10.006

Maybe authors will be interested in these works.

Author Response

Thank you for pointing the additional references, we have incorporated these as follows: “As a further instance, binuclear Cu(II) tetraazacyclotetradecane units show affinity towards carboxylates, which could be a further interesting option for a CO2 reduction strategy [27-31].”

Reviewer 2 Report

This paper aims at modelling the reduction ability of macrocycle nickel complexes towards carbon dioxide. Several methodologies were used to compare hydrogenated macrocycles with non-hydrogenated ones, the main conclusion being that the former will be promising candidates to CO2 fixation and reduction. The methodologies seem to be carried out properly although some points need to be addressed:

1 – Calculating enthalpies requires a full Hessian calculation to compute the normal modes. How do you address the imaginary frequencies of the fragments when calculating the interaction enthalpy?

2 – QTAIM BCP. The authors failed to indicate if this theory predicts (or not) a Ni-CO2 BCP in the complexes where the distances are outside the normal range of a covalent bond. The some applies to the Ni-N3 and Ni-N7.

3 – Looking at the pictures and angles seems to me that the complexes fall in two categories: i) square pyramidal complexes and ii) trigonal bipyramidal ones. The most promising one is of the later type with the CO2 occupying one of the Y positions. It would be interesting to compare these two possible geometries in the light of the old-fashioned Ligand Field Theory.

4 – The authors should provide HOMO values (in eV) for all the complexes since they may be indicative of the reduction capabilities of the metal center.

MAJOR POINT

5  - Hydrogenation makes every carbon in the macrocycle a chiral center with R and S conformations. I wonder if the authors tested all (or at least the most dramatic) possible conformations of the macrocycle. These conformations (may) have dramatic implications in the location of the N atoms. It is possible that other conformations can be more stable and more favorable (or not) to CO2 reduction. It may have happened that the most promising complex studied felt in the right conformation for CO2 reduction by serendipity, but the others may also have such a conformation. Information on this point must be provided to have a final decision on the manuscript which for the moment is Major Revision.

Author Response

This paper aims at modelling the reduction ability of macrocycle nickel complexes towards carbon dioxide. Several methodologies were used to compare hydrogenated macrocycles with non-hydrogenated ones, the main conclusion being that the former will be promising candidates to CO2 fixation and reduction. The methodologies seem to be carried out properly although some points need to be addressed:

1 – Calculating enthalpies requires a full Hessian calculation to compute the normal modes. How do you address the imaginary frequencies of the fragments when calculating the interaction enthalpy?

Response: There were no imaginary frequencies. Even the three (3N-6) frequencies for the bent CO2 structures are positive in the given ZPE correction.

2 – QTAIM BCP. The authors failed to indicate if this theory predicts (or not) a Ni-CO2 BCP in the complexes where the distances are outside the normal range of a covalent bond. The some applies to the Ni-N3 and Ni-N7.

Response: We are being more explicit now about the existence of Ni-N3 and Ni-N7 BCPs in the text. In addition for 1[NiL-CO2]0, the Ni-CCO2 BCP is found indeed and this fact is commented as follows at the end of the QTAIM BCP Analysis section 2.5: “In the case of 1[NiL-CO2]0, the Ni-CCO2 BCP has very low ρ, Δρ and DI values (0.003, 0.010 and 0.008, respectively) and the outcome of the QTAIM analysis when comparing the NiL unit in 1[NiL-CO2]0 and 1[NiL]0 is the same, not shown.”

3 – Looking at the pictures and angles seems to me that the complexes fall in two categories: i) square pyramidal complexes and ii) trigonal bipyramidal ones. The most promising one is of the later type with the CO2 occupying one of the Y positions. It would be interesting to compare these two possible geometries in the light of the old-fashioned Ligand Field Theory.

Response: Thank you for this comment. 3[NiL-CO2]2– is square-planar, 2[NiLH-CO2]+ can be considered as square-pyramidal, while 1[NiLH-CO2]0 has a distorted geometry. The d populations of all the complexes are collected in Table 3, according to the chosen coordination system (z-axis defines the Ni-CCO2 bond and the Ni-N(3) bond defines the xz-plane for 1[NiLH-CO2]0).

4 – The authors should provide HOMO values (in eV) for all the complexes since they may be indicative of the reduction capabilities of the metal center.

Answer: The MO eigenvalues (in au) are quoted in the ESI Tables S2a and S2b.

MAJOR POINT

5 - Hydrogenation makes every carbon in the macrocycle a chiral center with R and S conformations. I wonder if the authors tested all (or at least the most dramatic) possible conformations of the macrocycle. These conformations (may) have dramatic implications in the location of the N atoms. It is possible that other conformations can be more stable and more favorable (or not) to CO2 reduction. It may have happened that the most promising complex studied felt in the right conformation for CO2 reduction by serendipity, but the others may also have such a conformation. Information on this point must be provided to have a final decision on the manuscript which for the moment is Major Revision.

Response: Thank you for this comment. We have recalculated the NiLH compounds class and the text and results have been edited accordingly. We have checked also for few possible orientations on the chiral carbon centers to explore the energetics of 1[NiLH]2+, 2[NiLH]+ and 2[NiLH-CO2]+, see Table S1, to explore a possible space of 27 for carbons and include also a possible 28 option hydrogen orientation of nitrogens was not targeted.

Reviewer 3 Report

Present manuscript describes DFT investigation of CO2 binding affinity of series of Ni complexes. The research potentially could be interesting for the readership of the Molecules. However, the main disadvantage of this work is the lack of comparison of the obtained theoretical data on CO2 affinity for Ni complexes with experimental characteristics. In addition, there is no explanation of the choice of the calculation method, its validation or comparison with those used in the literature for calculating the electronic properties of nickel complexes. The authors did not conduct a sufficient and comprehensive analysis of the literature.

There are some concerns that need to be addressed:

1) The authors did not conduct a sufficient and comprehensive analysis of the literature, most of the cited literature is quite old. The choice of the complexes of investigation should be explained taking into account not only the previous works of the authors (ref 44, 45), but also other existing in the literature catalytic systems based on nickel complexes:

e.g. see

10.1016/j.ccr.2017.12.009

10.1039/D0CS00218F

10.1021/acs.chemrev.8b00361

2) The detailed rationale for the choice of the calculation method (B3LYP/6-311G*) should be added. It is not clear why authors used 6-311G* basis set for Ni, instead commonly used LANL2DZ. Authors should provide the cartesian coordinate of optimized geometries in SI or in separate XYZ file.

3) The Title of the research article should be short, concise and clear.

4) All abbreviations should explain at first appearance in the text, moreover all abbreviation and the ligand structure in the Abstract should be given in details. The designations used 1[Ni0LH]0, 2[NiILH]+ and 3[NiL]2− are not clear. MPA abbreviation which stands for Mulliken population analysis are not introduced in the text.

5) From the footnote of the Figure 1, it is not clear whether these are the structures of the complexes from X-ray or their DFT-optimized (B3LYP/6-311G*)

6) Figure 2: the colour scheme should be added as legenda or as footnote.

Author Response

Present manuscript describes DFT investigation of CO2 binding affinity of series of Ni complexes. The research potentially could be interesting for the readership of the Molecules. However, the main disadvantage of this work is the lack of comparison of the obtained theoretical data on CO2 affinity for Ni complexes with experimental characteristics. In addition, there is no explanation of the choice of the calculation method, its validation or comparison with those used in the literature for calculating the electronic properties of nickel complexes. The authors did not conduct a sufficient and comprehensive analysis of the literature.

There are some concerns that need to be addressed:
1) The authors did not conduct a sufficient and comprehensive analysis of the literature, most of the cited literature is quite old. The choice of the complexes of investigation should be explained considering not only the previous works of the authors (ref 44, 45), but also other existing in the literature catalytic systems based on nickel complexes:
e.g. see
10.1016/j.ccr.2017.12.009
10.1039/D0CS00218F
10.1021/acs.chemrev.8b00361

Response: Thank you for pointing out the additional important reviews, we are sorry to have missed these in our previous version. The suggested references are being cited within the Introduction section and further relevant comments with respect to differences in related electrocatalytic potentials have been considered.

2) The detailed rationale for the choice of the calculation method (B3LYP/6-311G*) should be added. It is not clear why authors used 6-311G* basis set for Ni, instead commonly used LANL2DZ. Authors should provide the cartesian coordinate of optimized geometries in SI or in separate XYZ file.

Response: 6-311G* basis set (including Wachters+f for Ni) is of triple-zeta quality while LANL2DZ basis set is a double-zeta one, i.e. 6-311G* (H 3s; C,N,O: 4s3p1d, S: 6s5p1d; Ni 8s6p4d1f) LANL2DZ (H 2s; C,N,O: 3s2p, S: 2s2p+core[10 electrons 2s1p]; Ni 3s3p2d+core[10 electrons 2s1p]). We prefer the usage of a triple zeta quality with the polarization functions included for the non-hydrogen atoms in our calculations.
The choice of a functional (or method) is a matter on its own. Nevertheless, we have tested other functionals as well, where the NiII complexes did not bind CO2, but the outcome was fairly the same, i.e. BLYP or B3LYP-GD3 (or even B2PLYP) confirmed that these complexes show no affinity towards CO2.

3) The Title of the research article should be short, concise and clear.

Answer: Thank you for this suggestion, the title has been modified as follows: “Ni oxidation state and ligand saturation impact on the capability of octaazamacrocyclic complexes to bind and reduce CO2”.

4) All abbreviations should explain at first appearance in the text, moreover all abbreviation and the ligand structure in the Abstract should be given in details. The designations used 1[Ni0LH]0, 2[NiILH]+ and 3[NiL]2− are not clear. MPA abbreviation which stands for Mulliken population analysis are not introduced in the text.

Answer: Thank you for the comment. The following sentence has been introduced into the results section 2.1. Energetics: “To further distinguish and characterize the complexes studied, the following notation is applied where appropriate: e.g. 2[NiILH]+ denotes the [NiLH] species in a doublet spin state, Ni in oxidation state I and a total charge of +1.”
MPA abbreviation is introduced at its’ first occurrence in the Introduction section.

5) From the footnote of the Figure 1, it is not clear whether these are the structures of the complexes from X-ray or their DFT-optimized (B3LYP/6-311G*)

Answer: Amended.

6) Figure 2: the colour scheme should be added as legenda or as footnote.

Answer: Amended.

Round 2

Reviewer 2 Report

In my opinion this paper does not fulfil the standards of a Q1 Journal in what concerns impact and soundness of the procedures adopted.

The questions risen in round one of reviewing were answered in an unconvincing way, although improving the manuscript.

1 - Not having an imaginary frequency in the 3N-6 coordinates is not the only criteria for an appropriate thermodynamics analysis: the other 6 should be zero or near zero and (the most important) the gradient must be zero. the bent geometries are surely having large non-zero gradient components.

If the authors should go ahead with the paper in this format they should provide a caution advice about a thermodynamic analysis carried out outside a minimum or saddle point.

2 - I'm satisfied with the answer

3 - d populations are not very useful for an experimentalist. My suggestion was  to compare energetically the d orbital splitting in the fields of the two extreme geometries. Sometimes that alone explains the differences in stabilities in a way an synthetic inorganic chemist can understand.

4 - in the same spirit of point 3 Hartree is not a unit you should use if you want to make your paper useful for the community of experimentalist. eV is the units to use for MO energies and kJ/mol in the energetics. But its an option of the authors.

5 - As reader I cant understand table S1 and I dont see how a reader can figure our that you tested 27 conformers neither the major differences between stability and structure among them.

In my opinion the paper still needs improvements to keep to the standards of a Q1 Journal. 

Author Response

Please find the response to the reviewer's comments in the attached pdf file.

Response to Reviewer 2.

Thank you for the additional comments in this second review round. Please find enclosed a point by point answer to the reviewers' queries.

1 - Not having an imaginary frequency in the 3N-6 coordinates is not the only criteria for an appropriate thermodynamics analysis: the other 6 should be zero or near zero and (the most important) the gradient must be zero. the bent geometries are surely having large non-zero gradient components.
If the authors should go ahead with the paper in this format they should provide a caution advice about a thermodynamic analysis carried out outside a minimum or saddle point.

Response: We give right to the reviewer. In accord with his suggestion, the interaction enthalpies are entirely excluded from the manuscript.

3 - d populations are not very useful for an experimentalist. My suggestion was to compare energetically the d orbital splitting in the fields of the two extreme geometries. Sometimes that alone explains the differences in stabilities in a way a synthetic inorganic chemist can understand.

Response: In this study, we focused on existing structures with respect to the optimized geometries. This comment is relevant for further investigation of the impact of coordination polyhedron (including charge and multiplicity, etc.) on the affinity of Ni complexes to bind CO2. However, this is far beyond the scope
of the present manuscript. Here, we focus on d-populations, because these can be correlated with the oxidation state change, when CO2 is bound to Ni in a given complex.

4 - in the same spirit of point 3 Hartree is not a unit you should use if you want to make your paper useful for the community of experimentalist. eV is the units to use for MO energies and kJ/mol in the energetics. But its an option of the authors.

Response: Amended, Hartree unit is replaced by eV now.

5 - As reader I cant understand table S1 and I dont see how a reader can figure our that you tested 27 conformers neither the major differences between stability and structure among them.

Response: Table S1 is now supported by an extra zip file containing all 27 structures. We would like to emphasize that the reported geometries in Table S1 are minima on the potential energy surface (confirmed via the vibrational analysis) We have tested four additional conformers of 1[NiLH]2+ without CO2 and two for 2[NiLH]1+ with and without CO2. upon your well pointed request in the previous revision round. The stability between these structures was resolved by the comparison of their total energies as summarized in Table S1. I am sorry to have forgotten to upload an xyz zip file of all geometries in Table S1 in the previous revision round. This issue was also mentioned in the report of reviewer 3. Furthermore, the two initial structures are shown in Figure 1, five major structures with CO2 are shown in Figure 2 and additional three structures without CO2 are compared to those which bind CO2 in Figure S1. Information on the geometrical parameters (bond distances and/or angles) is compiled in Tables 1, S2a, S2b as well as S5a and S5b. Still, both reviewers are right that the missing xyz files made the paper difficult to follow.

In my opinion the paper still needs improvements to keep to the standards of a Q1 Journal.

Response: We hope to have appropriately addressed reviewer's concerns in this second review round.

Reviewer 3 Report

In general I am satisfied with revised version of the manuscript and authors reply. 

I have following remarks:

1) In the Table  1, the energy values are given with 3 significant digits, is it necessary?

2) Please provide XYZ file with Cartesian coordinates and energies of optimized geometries.

3) It would also be helpful ha have a table with the geometric characteristics of the reaction center (Ni-N distances, N-Ni-N angles, N-N-N-N dihedral angles).

Author Response

Please find the response to the reviewer's comments in the attached pdf file.

Response to Reviewer 3.
We have amended all the reviewers´ comments as follows:

1) In the Table 1, the energy values are given with 3 significant digits, is it necessary?

Response: Thank you for the comment, the energy difference values in Table 1 have been shortened to two
decimal places.

2) Please provide XYZ file with Cartesian coordinates and energies of optimized geometries.

Response: I apologize for the created inconvenience. The corresponding files have been uploaded now.

3) It would also be helpful to have a table with the geometric characteristics of the reaction center (Ni-N
distances, N-Ni-N angles, N-N-N-N dihedral angles).

Response: Ni-N distances, N-Ni-N angles and Ni_N-N angles are shown in Tables S2a and S2b. Further
detail can be found in the xyz files provided.
